# Offline Reinforcement Learning for Traffic Signal Control

## Abstract

Traffic signal control is an important problem in urban mobility with a significant potential of economic and environmental impact. While there is a growing interest in Reinforcement Learning (RL) for traffic signal control, the work so far has focussed on learning through simulations which could lead to inaccuracies due to simplifying assumptions. Instead, real experience data on traffic is available and could be exploited at minimal costs. Recent progress in *offline* or *batch* RL has enabled just that. Model-based offline RL methods, in particular, have been shown to generalize from the experience data much better than others.

We build a model-based learning framework which infers a Markov Decision Process (MDP) from a dataset collected using a cyclic traffic signal control policy that is both commonplace and easy to gather. The MDP is built with pessimistic costs to manage out-of-distribution scenarios using an adaptive shaping of rewards which is shown to provide better regularization compared to the prior related work in addition to being PAC-optimal. Our model is evaluated on a complex signalized roundabout showing that it is possible to build highly performant traffic control policies in a data efficient manner.

## 1 Introduction

Road traffic signal control has attracted substantial interest as an application of reinforcement learning (RL) (Wei et al., 2019; Yau et al., 2017). However most published work in the area is unlikely to be applied in practice as trial and error methods for interacting with the environment are not feasible in the real world. Similarly trying to infer an RL policy using a simulator does not take advantage of the fact that real experience data about traffic is available from transportation management operators.

A more appropriate and realistic set up is to use offline RL training to learn from static experience data (Lange et al., 2012). A typical data set will consist of a set of tuples of the form $\{s_i, a_i, r_i, s_{i+1}\}$, *i.e.*, when the system was in state $s_i$, action $a_i$ was taken, which resulted in a reward $r_i$ and the system then transitioned into a new state $s_{i+1}$. From the experience data the objective is to learn a policy, *i.e.*, a mapping from state to action which maximizes the long term expected cumulative reward. In a traffic signal control setting, the state captures the distribution of traffic on the road network, the action space consists of different phases (red, green, amber) on signalized intersections and the reward is a metric of traffic efficiency.

Compared to online RL approaches (Sutton & Barto, 2018), offline (or batch) RL shifts the focus of learning from data exploration to data-driven policy building. The offline policy building is challenging due to deviation in the state-action visitation by the policy being learned and the policy that logged the static dataset (Fujimoto et al., 2019b; Levine et al., 2020). A number of different solution frameworks are proposed for offline RL that are either model-free (Fujimoto & Gu, 2021; Kumar et al., 2020; Wu et al., 2019; Kostrikov et al., 2021b;a) or derive a Markov Decision Process (MDP) model (Shrestha et al., 2020; Yu et al., 2020; Kidambi et al., 2020; Yu et al., 2021) from the data set. We focus on model-based RL approaches which have been shown to offer better regularization in presence of uncertain data. Such approaches are characterized by a mechanism to penalize under-explored or under-represented transitions, a notion referred to as *pessimism under uncertainty* (Fujimoto et al., 2019b). While the RL algorithms, by nature, are highly sensitive to hard-to-tune hyper-parameters (Ng, 2022), incorporation of the uncertainty penalties adds further supplementary hyper-parameters. These are shown to affect the performance of offline learning significantly (Lu et al.,

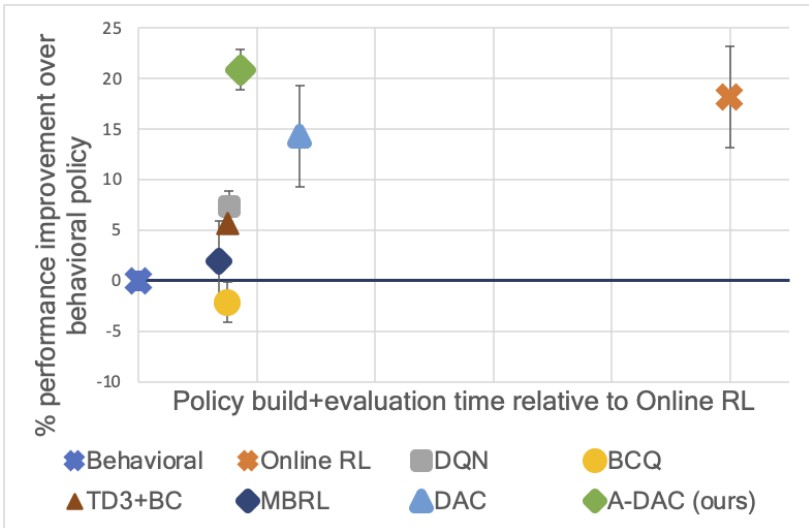

Figure 1: A-DAC, our model-based offline RL approach for traffic control achieves the best performance out-of-the-box compared to three model-free (DQN (Mnih et al., 2016), BCQ (Fujimoto et al., 2019b), and TD3+BC (Fujimoto & Gu, 2021)) , one model-based RL (MBRL (Yu et al., 2020)), and our predecessor (DAC (Shrestha et al., 2020)) RL, predecessor of A-DAC. The dataset is collected by a behavioral policy that cycles through each traffic signal change action. DQN is greedy and prone to exploration errors. BCQ, TD3+BC and MBRL fail to generalize despite their built-in pessimism. DAC is sensitive to its hyperparameters and needs multiple online evaluations. An Online RL baseline is included for comparison which is matched or bettered by our approach in a fraction of time.

2021). Since interaction to the environment is not permitted in the offline setting, adjustments to the hyper-parameters are not trivial.

We build on the DAC framework (Shrestha et al., 2020), which derives a finite Markov Decision Process (MDP) (Puterman, 2014) from a static dataset and solves it using an optimal planner. The MDP derivation uses empirical averages to interpolate contributions from nearby transitions seen in the dataset with pessimistic penalties applied to derivation of the rewards based on the coverage of the neighborhood. DAC, in a manner similar to the model-based offline RL algorithms discussed previously, is sensitive to its hyper-parameters relating to the pessimistic costs. We incorporate an adaptive pessimistic reward shaping in DAC which makes it both robust to dataset properties and significantly faster to train by eliminating online interactions required for tuning magnitude of conservatism. Our adaptive reward penalty formulation relies on local Lipschitz smoothness (O'Searcoid, 2006) assumption on the $Q$-values exhibited by the true (but unknown) MDP, a much weaker assumption compared to the DAC-MDP framework. We provide the same theoretical guarantees on optimality while exhibiting a superior empirical performance.

Figure 1 illustrates our contribution, dubbed Adaptive(A)-DAC, evaluated on a real traffic signal control setup. A-DAC finds a policy significantly better than a common behavioral (or data collection) policy in a small fraction of time compared to online learning and does so out-of-the-box. A key insight of our approach is that data collected from cyclic policies that are oblivious to rate of traffic arrival and is often the norm in many traffic signal scenarios, can be leveraged to infer superior policies which improve overall traffic efficiency.

**Contributions:**

- We formulate traffic signal control as an offline RL problem. While RL has recently been proposed for offline optimization, to the best of our knowledge, it has not been used for the traffic signal control before.

- We extend a recent model-based offline RL framework, DAC, to our problem and improve it by employing an adaptive reward penalty mechanism that enables the best trade-offs in the performance

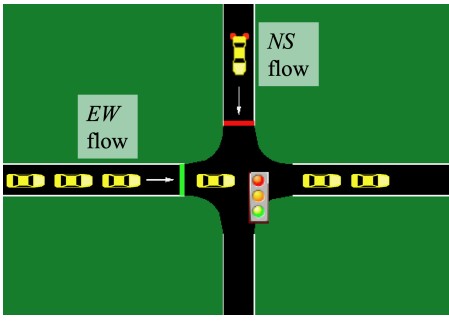

Figure 2: A simple signalized intersection with two traffic flows: North-South ($NS$) and West-East ($EW$).

Table 1: A small experience dataset collected using *Cyclic* traffic signal control policy applied on the intersection in Figure 2.

| State $s$ ($|NS|, |EW|$) | Action $a$ | State $s'$ | Reward $r$ |
|---|---|---|---|
| (1,5) | $EW$ | (3,3) | 2 |
| (3,3) | $NS$ | (1,5) | 2 |
| (6,1) | $NS$ | (2,3) | 4 |
| (2,3) | $EW$ | (6,1) | 2 |
| (0,5) | $EW$ | (2,3) | 2 |
| (2,3) | $NS$ | (0,5) | 2 |

Table 2: Derived rewards for the core states identified from the experience dataset in Table 1.

| State $s$ ($|NS|, |EW|$) | Averagers ($C = 0$) | | DAC @ $C = 1$ | | DAC @ $C = 2$ | | A-DAC | |
|---|---|---|---|---|---|---|---|---|
| | $R(s, NS)$ | $R(s, EW)$ | $R(s, NS)$ | $R(s, EW)$ | $R(s, NS)$ | $R(s, EW)$ | $R(s, NS)$ | $R(s, EW)$ |
| (2,3) | 2.67 | 2 | 2.41 | 1.77 | 1.85 | 1.25 | 1.58 | 1.53 |
| (6,1) | 2.67 | 2 | 2.29 | 1.16 | 1.46 | -0.7 | 1.17 | 0.32 |
| (3,3) | 2.67 | 2 | 2.45 | 1.66 | 1.98 | 0.89 | 1.82 | 1.31 |
| (1,5) | 2.67 | 2 | 2.14 | 1.85 | 0.96 | 1.52 | 0.55 | 1.70 |
| (0,5) | 2.67 | 2 | 2.03 | 1.82 | 0.63 | 1.43 | 0.14 | 1.65 |

and the policy building overheads. We provide Probably-Approximately-Correct (PAC) guarantees for A-DAC under more relaxed and realistic assumptions on the Q-function.

- We propose a methodology for data collection at a traffic intersection using macro statistics provided by traffic authorities. We evaluate our approach on a complex signalized roundabout where traffic is coordinated using eleven phases.

**Outline:** The rest of the paper is structured as follows. In Section 2, we introduce a small idealized traffic control problem as a working example. A primer on offline RL methods and their applicability to traffic control is provided in Section 3. In Section 4, our proposed approach, A-DAC, is introduced and elaborated upon. Section 5 reasons about suitability of our approach to the traffic control problem. Section 6, then, evaluates A-DAC and other baselines using a complex signalized roundabout. The most relevant related work is presented in Section 7 and we conclude in Section 8 with a summary of the work.

## 2  Basic Traffic Signal Control

We start with a simple scenario of traffic signal control first presented in Rizzo et al. (2019) where an optimal model for traffic light phase duration is derived based on simplistic assumptions. When these assumptions are relaxed in a realistic setting, more complex machinery is required for optimal control which we present later.

Consider a traffic signal at an intersection which controls traffic in only two directions: either from north to south ($NS$) or from west to east ($EW$) (See Figure 2). Suppose the traffic follows a Poisson process with the rate of traffic arrival being $\lambda_1$ and $\lambda_2$ respectively for the two traffic flows. The traffic starts arriving from time $t = 0$ and the total cycle time is $T$. The number of vehicles entering any incoming traffic arms is uniformly distributed by definition of the Poisson process and the expected number at time $t$ is given by $\lambda t$. The optimal setting for the duration of the green phase for $NS$ can be derived by minimizing the average delay faced by a vehicle. It evaluates to: $\frac{\lambda_1}{\lambda_1 + \lambda_2} T$. The equation states that the green phase duration should be proportional to the arrival rate of the vehicles. It can be easily generalized to a case of $n$ exclusive traffic flows where the optimal green phase duration for an edge $i$ is $\frac{\lambda_i}{\sum_{i=0}^{n} \lambda_i} T$.

We relax the assumption of the known arrival rate. Instead, we assume an experience dataset of vehicle movement at the intersection exists where the signal is controlled by a cyclic policy. The *Cyclic* policy simply alternates green phase between the *NS* flow and the *EW* flow at every time step, mimicking a commonly observed scenario. The idea is to use the dataset to devise a smarter control policy. We study some of the recent developments in offline RL towards this problem scenario before introducing a real signalized roundabout we target. The techniques we develop are general enough to apply directly to the more complex test environment.

## 3 Offline Reinforcement Learning

In Reinforcement Learning (RL) (Sutton & Barto, 2018), an agent interacts with an environment, assumed to be a Markov Decision Process (MDP) (Puterman, 2014), in order to learn an optimal control (action selection) policy. The MDP is given by a tuple $(\mathcal{S}, \mathcal{A}, P, R, \gamma)$, with a state space $\mathcal{S}$, an action space $\mathcal{A}$, transition dynamics $P(s, a, s')$, a stochastic reward function $R(s, a)$ and a discount factor $\gamma \in [0, 1)$. The agent aims to learn a policy function $\pi : \mathcal{S} \to \mathcal{A}$ which maximizes the expected sum of discounted rewards. Formally, the objective of RL is given by the following.

$$\max_{\pi} \mathbb{E}_{\substack{a_t \sim \pi(.|s_t) \\ s_{t+1} \sim P(.|s_t, a_t)}} \Big[ \sum_{t=0}^{\infty} \gamma^t R(s_t, a_t) \Big] \tag{1}$$

The policy $\pi$ has a Q-function $Q^\pi(s, a)$ giving the expected infinite-horizon discounted reward starting with state-action pair $(s, a)$. The optimal policy $\pi^*$ maximizes the Q-function over all policies and state-action pairs. The maximum Q-values are computed by repeated application of the Bellman backup (Bellman, 1966) operator $B$ stated below.

$$B[Q](s, a) = R(s, a) + \gamma \mathbb{E}_{s' \sim P(.|s, a)} \Big[ \max_a Q(s', a) \Big] \tag{2}$$

RL strives to discover a near-optimal policy by exploring actions in the environment. In an offline or batch setting (Levine et al., 2020; Ernst et al., 2005), the environment is replaced by a static dataset collected apriori. The dataset $\mathcal{D}$ is made up of tuples $\{(s_i, a_i, r_i, s'_i)\}$ where each tuple takes action $a_i$ from state $s_i$ to transition to state $s'_i$ while giving the reward $r_i$. The dataset is collected from multiple episodes/trajectories of the form $(s_1, a_1, r_1, s_2, \ldots, s_H)$ where $H$ is the trajectory length.

*Example:* *Table 1 presents a sample experience data set collected by a policy cycling between the actions NS and EW. The state is given by a vector of the number of vehicles arriving from each incoming lane. The data set includes three trajectories, each contributing two tuples. Duplicates are removed to make the example concise.*

In the basic traffic control setup outlined in Section 2, an action corresponds to activating green phase for one of the traffic flows for a fixed time duration, called an *observation period*. Therefore, $a_i \in \{NS, EW\}$. Observation state is given by a vector of the number of vehicles arriving from each incoming traffic arm, making it 2-dimensional in our case. Reward is a non-negative integer denoting the number of vehicles that cross the signal during the observation period.

### 3.1 Model-free Learning

The first solution approach we consider is to adapt a popular *off-policy* Q-learning approach Deep Q Network (DQN) (Mnih et al., 2015) to the offline setting. The offline setting often causes extrapolation errors in Q-learning which result from mismatch between the dataset and the state-action visitation of the policy under training. Fujimoto et al. (2019b) proposes a batch-constrained Q learning (BCQ) approach to minimize distance between the selected actions and the dataset. BCQ trains a state-conditioned generative model of the experience dataset. In discrete settings, the model $G_\omega$ gives a good estimate of the behavioral policy $\pi_b$ used to collect data. That means, we can constrain the selected actions to data using a threshold $\tau \in [0, 1)$:

$$\pi(a|s) = \underset{a' | G_\omega(a'|s) / \max_{\hat{a}} G_\omega(\hat{a}|s) > \tau}{\arg \max} Q(s, a') \tag{3}$$

While the BCQ is effective at pruning the under-explored transitions, its benefits are limited when the behavioral policy tends to a uniform distribution which holds true in our case: The behavior policy $\pi_b$ we aim to utilize is *Cyclic* for which $\pi_b(a|s) = \pi_b(a) = \frac{1}{|\mathcal{A}|}$.

In another offline RL approach, a minimalistic change to classic TD3 algorithm (Fujimoto et al., 2018) is proposed in (Fujimoto & Gu, 2021) which regularizes the TD3 policy with a behavioral cloning (BC) (Osa et al., 2018; Pomerleau, 1988) term:

$$\pi \leftarrow \arg\max_\pi \mathbb{E}_{(s,a)\sim D} \left[ \lambda \, Q\big(s, \pi(s)\big) - \big(\pi(s) - a\big)^2 \right] \tag{4}$$

Here, $\lambda$ is a normalizing scalar that is set to a value inverse of the average critic ($Q$) function value. This approach is termed TD3+BC and is regarded as the current state-of-the-art model-free offline RL algorithm.

## 3.2 Derived MDP-based Learning

A contrastive approach to constraining the RL to the dataset is to derive an MDP from the data and either solve it optimally or use model-based policy optimization (MBPO) (Janner et al., 2019). This approach provides better generalization since each transition gets more supervision compared to the off-policy approaches. There exist multiple recent approaches built on this principle called Model-based (MB-)RL (Sutton, 1991). MBRL primarily learns an approximate transition model and (optionally) a reward model by supervising data followed by a phase of uncertainty quantification to deal with out-of-distribution visitations (Yu et al., 2020; Kidambi et al., 2020; Yu et al., 2021). We employ a simple instantiation, called DAC-MDP (Shrestha et al., 2020), which is based on Averagers' framework (Gordon, 1995). The idea is to learn transitions and rewards as empirical averages over the nearest neighbors in the state-action space. It lends to a natural approximation and enables an intuitive uncertainty quantifier. Furthermore, DAC-MDP creates a finitely-represented MDP by working on a set of *core* states which can be reached from any *non-core* state using a one-step transition and do not allow transition to a non-core state from within. The finite structure of the MDP implies that an accurate solution is feasible (such as by using a value iteration solver) despite an infinite continuous state space. We contribute an adaptive reward penalty mechanism to the DAC framework which works as an effective uncertainty quantifier. Before outlining our contributions, we formalize the DAC framework.

**Assumption 3.1.** We assume a continuous state space $\mathcal{S}$ and a finite action space $\mathcal{A}$. We are given a nearest neighbor function $NN(s, a, k, \alpha)$ that finds at most $k$ nearest neighbors to a state-action pair $(s, a)$ with an optional maximum distance threshold $\alpha$. $NN$ uses a metric function $d$, such as *Euclidean* (Toth et al., 2017), that keeps the pairs with different actions infinitely distant while the distance between the pairs with the same action is evaluated on the state metric space: For example, $d(s_i, a_i, s_j, a_j) = ||s_i - s_j||$ if $a_i = a_j$, $\infty$ otherwise.

We use shorthand $d_{ij}$ for distance between pairs $(s_i, a_i)$ and $(s_j, a_j)$. Notation $d'_{ij}$ indicates a normalized version of distance $d_{ij}$. Given a smoothness parameter $k$ and a distance threshold $\alpha$, the derived MDP $\tilde{M}$ is defined below.

**Definition 3.2.** MDP $\tilde{M} = (\mathcal{S}, \mathcal{A}, \tilde{R}, \tilde{P}, \gamma)$ shares the state space and the action space of the underlying MDP $M$. For a state-action pair $(s, a)$, let $kNN = NN(s, a, k, \alpha)$ be its nearest neighbors from $\mathcal{D}$. The reward and transition functions are then defined as:

$$\tilde{R}(s, a) = \frac{1}{k} \sum_{i \in kNN} r_i, \;\; \tilde{P}(s, a, s') = \frac{1}{k} \sum_{i \in kNN} I[s' = s'_i]$$

DAC modifies the reward function to penalize transitions to an under-explored region with an additive penalty parameterized by a cost parameter $C \geq 0$:

$$\tilde{R}(s, a) = \frac{1}{k} \sum_{i \in kNN} \big(r_i - C * d(s, a, s_i, a_i)\big) \tag{5}$$

It should be noted that while the reward shaping in model-based online RL acts as a way to incorporate additional incentive based on domain knowledge to an otherwise sparse reward function (Ng et al., 1999), the offline setting uses it as a means to incorporate pessimism to the MDP.

***Example:*** *Table 2 compares the rewards derived with different cost penalties. $k = 3$ throughout and the distances are normalized to the maximum distance for ease of presentation. As an example, $\tilde{R}_{C=0}((2,3), NS) = \frac{1}{3}(r[1] + r[2] + r[5]) = \frac{1}{3}(4 + 2 + 2) = 2.67$, where $r[.]$ is indexes Table 1. State $(2, 3)$, for which the reward $r$ from dataset is equal for both actions, is assigned a higher reward for action NS due to influence from a high-reward neighbor $(6, 1)$.*

The DAC framework builds a finite MDP by focusing on a set of *core* states that are extracted from the data set $\mathcal{D}$ as $\mathcal{S}_{\mathcal{D}} = \{s'_i | (s_i, a_i, r_i, s'_i)\}$. As stated previously, states within $\mathcal{S}_{\mathcal{D}}$ do not transition to non-core states. The MDP built over the core states is solved using any standard tabular solver such as value iteration to compute values $\tilde{V}$ for the core states. We can then compute $\tilde{Q}$ for a non-core state using the following 1-step lookup which is used to transition from the non-core state to one of the core states:

$$\tilde{Q}(s,a) = \tilde{R}(s,a) + \gamma \sum_{s' \in \tilde{P}(s,a)} \tilde{P}(s,a,s').\tilde{V}(s') \tag{6}$$

***Example:*** *Consider a new state $(1, 4)$. Its Q-values on MDP with no cost penalties and $\gamma = 0.99$ evaluate to $\tilde{Q}_{C=0}((1,4), NS) = 7.92$ and $\tilde{Q}_{C=0}((1,4), EW) = 7.25$. The policy $\tilde{\pi}$ would, therefore, choose action NS which is suboptimal since the state indicates more traffic on the EW lane. Cost penalties help us in this instance: The MDP derived with $C = 2$ gives $\tilde{Q}_{C=2}((1,4), NS) = 32.6$ and $\tilde{Q}_{C=2}((1,4), EW) = 32.9$. The setting $C = 2$ is not arbitrary: it is the lowest $C$ value for which the EW action gets chosen. We discuss this choice further in the next section.*

## 4  A-DAC MDP Derivation

We have just shown through example that building offline solutions for traffic control is not trivial even in the simplest of the scenarios. We build an adaptive approach for reward shaping that retains the optimality guarantees while improving the efficiency of the DAC framework significantly; the resulting framework is called as A-DAC.

**Definition 4.1.** A-DAC automatically adjusts the penalties built into the reward function of the derived MDP $\tilde{M}$ in the following manner:

$$\text{Given } kNN = NN(s, a, k, \alpha) \text{ and } r_{max} = \max_{i \in kNN} r_i$$

$$\tilde{R}(s,a) = \frac{1}{k} \sum_{i \in kNN} r_i - r_{max} * d'(s, a, s_i, a_i)$$

It should be noted that $d'$ is a normalized version of $d$ which brings the penalty term to the units of rewards.[1] The intuition behind using the max reward in the neighborhood is to penalize the under-explored but highly rewarding transitions more heavily. Let's understand with an example.

***Example:*** *We saw earlier that for state $(2, 3)$ from Table 2, action NS brings a higher reward in DAC. This is due to influence from a high-reward-getter neighbor $(6, 1)$. It can be noticed from the same table that A-DAC's adaptive rewards make both the actions equally rewarding. Moreover, Q-values for a non-core state $(1, 4)$ evaluate to $\tilde{Q}_{A\text{-}DAC}((1,4), 'NS') = 31.4$ and $\tilde{Q}_{A\text{-}DAC}((1,4), 'EW') = 32.5$ selecting the action EW automatically.*

We present a canonical use case in Figure 3 to illustrate how the rewards are shaped in A-DAC. Of the $k$ neighbors considered, one neighbor is kept floating to simulate different types of neighborhood. e.g. $r_{max} = 1$

---

[1]For practicality, we use max-normalization: $d' = d/d_{max}$, where $d_{max}$ is the diameter of the point space and is approximately calculated by sampling a few points from data set $\mathcal{D}$ and aggregating distance from these points to all points in $\mathcal{D}$.

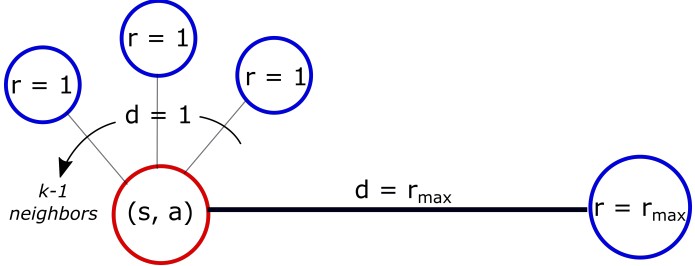

Figure 3: A configuration of $k$-nearest neighbors for state $(s, a)$ where $k - 1$ neighbors are at distance 1 each with reward 1. The remaining neighbor floats at distance $r_{max}$ bringing in reward $r_{max}$ where $r_{max} > 1$.

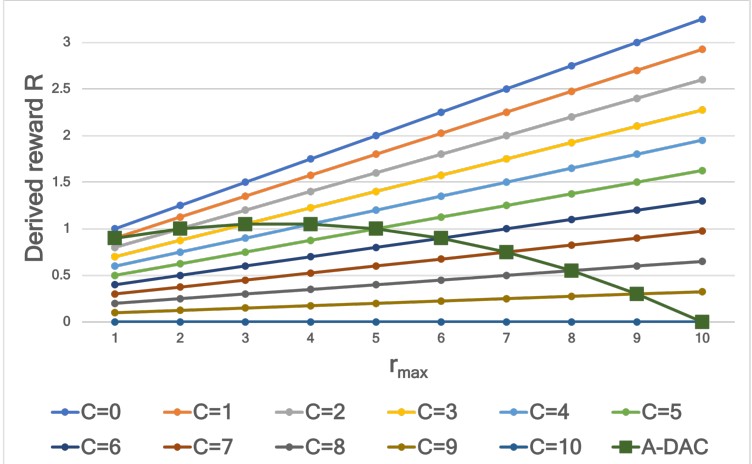

Figure 4: Comparison of rewards derived by DAC with different settings of cost $C$ to those obtained by A-DAC using the configuration in Figure 3 controlled by variable $r_{max}$. A-DAC penalizes the configurations with under-explored regions more while keeping the rewards high for homogeneous configurations.

gives the most homogeneous configuration, while a high $r_{max}$ replicates an under-represented region. Figure 4 shows how a global cost parameter $C$ would shape the reward for $(s, a)$. While a low $C$ gives high rewards for an under-represented region, a high $C$ is detrimental to the homogeneous configurations. A-DAC can be seen to offer a good balance.

**Optimality.** The policy learned by solving the A-DAC $\tilde{M}$, denoted $\tilde{\pi}$, can potentially be arbitrarily sub-optimal in true MDP $M$. We obtain a lower bound on the values obtained by policy $V^{\tilde{\pi}}$ in relation to the values $V^*$ provided by the optimal policy $\pi^*$ in $M$ under a "smoothness" assumption.

**Assumption 4.2.** A-DAC assumes local Lipschitz continuity (O'Searcoid, 2006) for Q-function $Q$: For a state-action pair $(s_i, a_i)$ and another pair $(s_j, a_j)$ in its neighborhood, *i.e.*, $d'_{ij} < \alpha$ for $\alpha \in [0, 1]$, there exists a local constant $L_Q(i, \alpha) \geq 0$ such that $|Q(s_i, a_i) - Q(s_j, a_j)| \leq L_Q(i, \alpha) d'_{ij}$.

The local continuity is a much weaker assumption compared to the global smoothness assumed in the DAC framework. Further, it is found to be practical based on an analysis of our traffic control setup presented in Section 5. In addition, we assume that the rewards are bounded to $[0, R_{max}]$ which holds for most practical applications including ours. $\tilde{\pi}$, then, provides the following PAC guarantee.

**Theorem 4.3.** *Given a static dataset $\mathcal{D}$ with its sample complexity indicated by covering number (see Definition A.2) $\mathcal{N}_{\mathcal{SA}}(\alpha)$ and an A-DAC MDP $\tilde{M}$ built on $\mathcal{D}$ with parameters $k$ and $\alpha$, if $\frac{\tilde{Q}_{max}^2}{\epsilon_s^2} \ln\left(\frac{2\mathcal{N}_{\mathcal{SA}}(\alpha)}{\delta}\right) \leq k \leq \frac{2\mathcal{N}_{\mathcal{SA}}(\alpha)}{\delta}$, then*

$$V^{\tilde{\pi}} \geq V^* - \frac{2\epsilon_s + \bar{d}_{max} R_{max}}{1 - \gamma}, \ w.p. \ 1 - \delta$$

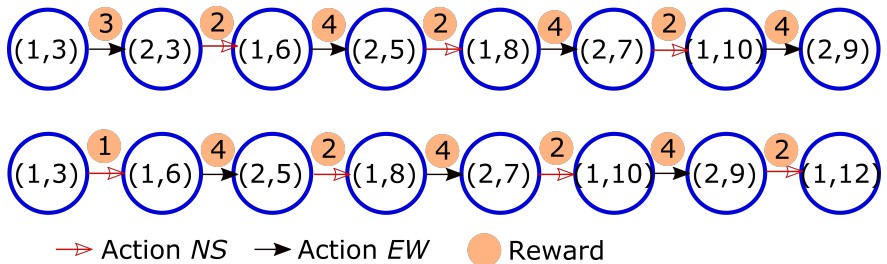

Figure 5: Two experience trajectories from the traffic intersection in Figure 2 taken using *Cyclic* behavior policy under a fixed traffic load assumption.

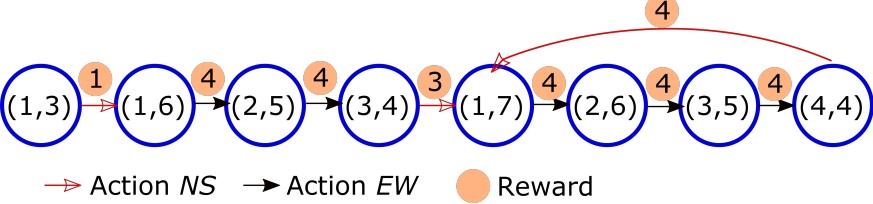

Figure 6: Policy learned by A-DAC using the dataset in Figure 5 improves the behavior policy by 33%.

The proof is presented in Appendix A. The first component $\epsilon_s$ denotes the maximum sampling error caused by using a finite number of neighbors; it helps us set a lower bound on $k$. The second component denotes the estimation error due to neighbors being at non-zero distance: Quantity $\bar{d}_{max}$ gives the worst case average distance which is upper bounded by $\alpha$ and can be lowered by augmenting the data set $\mathcal{D}$ with more diverse data.

## 5 A-DAC's Feasibility to Traffic Control

We use the simple two action single intersection model in Figure 2 to analyze A-DAC's feasibility to traffic control.

We assume a fixed rate of arrival for each of the two incoming lanes where $\lambda_{EW} = 3 * \lambda_{NS}$. The starting state is assumed to be $(NS, EW) = (1, 3)$. A maximum capacity $(R_{max})$ of 4 is allowed in order to maintain a steady flow. On this setup, two experience trajectories are collected using *Cyclic* control policy as illustrated in Figure 5. It can be easily seen that this policy moves $3T$ vehicles in $T$ timesteps. Next, A-DAC is trained on this toy dataset. Figure 6 shows a rollout with the same starting state. It can be noticed that a cumulative reward of $4T$ is achieved, giving 33% improvement over the behavioral policy (and matching the optimal policy). It shows that A-DAC generalizes well. Inspect state $(2,5)$ as an instance: The only action observed from it in the dataset is $NS$, but our policy has learned to take action $EW$ instead.

Finally, using the same model, we can show that for the Q-function, local Lipschitz continuity is the right condition to enforce. For example, suppose the arrival rate on each lane is $\lambda_1$ and $\lambda_2$ respectively. Assume, we start the cycle from the $\lambda_1$ lane, then the optimal return is:

$$J(\pi^*) = \frac{\lambda_1^2}{\lambda_1 + \lambda_2} + \gamma * \frac{\lambda_2^2}{\lambda_1 + \lambda_2} + \gamma^2 * \frac{\lambda_1^2}{\lambda_1 + \lambda_2} + \dots$$
$$= \frac{1}{1 - \gamma^2} * \frac{\lambda_1^2}{\lambda_1 + \lambda_2} + \frac{\gamma}{1 - \gamma^2} * \frac{\lambda_2^2}{\lambda_1 + \lambda_2}$$

Now if we assume that the $\lambda_1 > 0$ and $\lambda_2 > 0$ and $\lambda_1 + \lambda_1 \geq \delta$, the above expression is upper bounded by a convex function $c * (\lambda_1^2 + \lambda_1^2)$ for some $c > 0$ which is locally Lipschitz (as a function of $\lambda_1$ and $\lambda_2$).

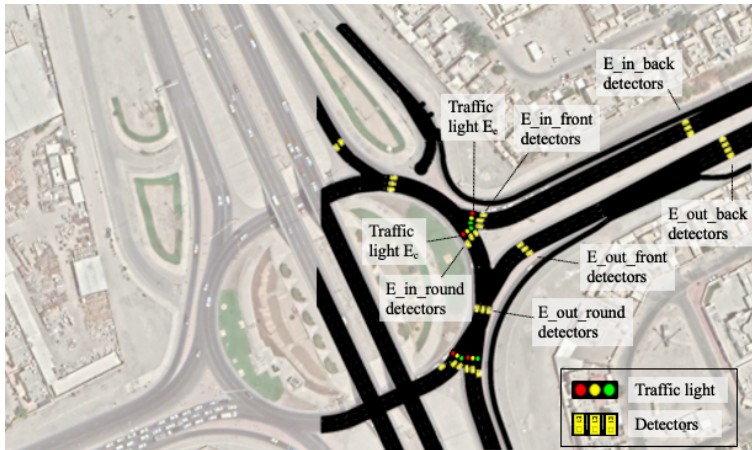

Figure 7: A signalized roundabout A-DAC optimizes. 68 loop detector devices installed in and around the junction report the traffic they observe which forms the state.

## 6 Evaluation

This section addresses the following questions:
*1. How well is the data collected from simple cyclic behavioral policy exploited by offline learners? (§6.3)*
*2. Does a partially trained behavior policy offer any added benefits to offline learning? (§6.4)*
*3. How do different latent state representations used in A-DAC compare? (§6.5)*
*4. How does A-DAC's efficiency and hyperparameter sensitivity compare to prior work? (§6.6)*

Before delving into these questions, we describe a real complex traffic roundabout environment used in evaluation (§6.1) and the details of our experimental setup (§6.2).

### 6.1 Environment for a Signalized Roundabout

We model a signalized roundabout shown in Figure 7. It is a complex intersection containing multiple lanes in each traffic arm and 10 traffic signals controlling the area. We model the state of traffic as a 68-dimensional vector, each dimension providing the number of vehicles seen at a designated location. An action corresponds to a phase of traffic that covers a certain configuration of the traffic signals. Details on engineering the states, actions, and rewards for this intersection are provided in Appendix B. The appendix also provides details on creating an experience batch from a micro-simulator bootstrapped with macro traffic statistics when access to the real experience trajectories data is limited.

### 6.2 Experimental Setup

We use the signalized roundabout detailed in Section 6.1 for evaluation. Each episode lasts an hour with 360 time steps. O-D matrix data is available for each hour of a typical weekday which allows us to create a single day batch. A random noise is added to the matrix when creating a larger batch. Two batches are used: (a) *1 day batch* giving ≈10k timesteps, and (b) *1 week batch* giving ≈70k timesteps.

**Behavioral policies.** We study two data collection policies:
1. *Cyclic*: Cycles through all actions. Represents a scenario commonly found across traffic intersections.
2. *Partial-RL*: An RL policy is partially trained via online interactions. A noisy version of the policy is then used to control data. This policy has been shown to be suitable for offline learning previously (Fujimoto et al., 2019a).

Batch collection and evaluation is carried out using SUMO microsimulator (Lopez et al., 2018). A fixed five hour workload, that corresponds to five RL episodes, is used for evaluation throughout: it includes two hours

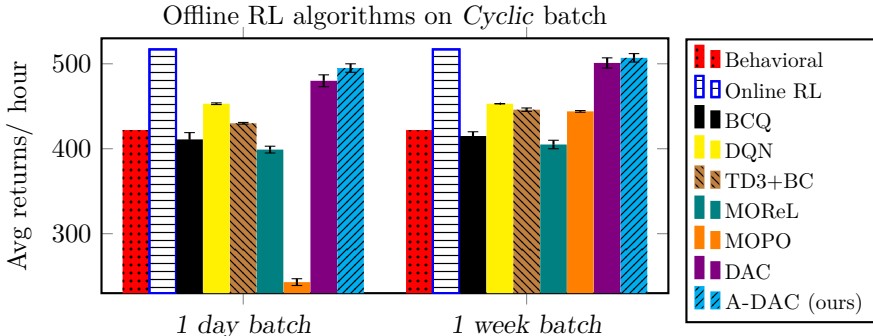

Figure 8: Comparison of offline RL algorithms on *Cyclic* batch. Error bars indicate the min-max interval obtained after 5 runs with different seeds under the best model settings. BCQ, TD3+BC, MOReL, and MOPO fail to improve the behavioral policy. Both DAC and A-DAC improve the policy significantly, even matching the Online RL performance with the larger batch, While DAC needs a large evaluation overhead, A-DAC works out-of-the-box.

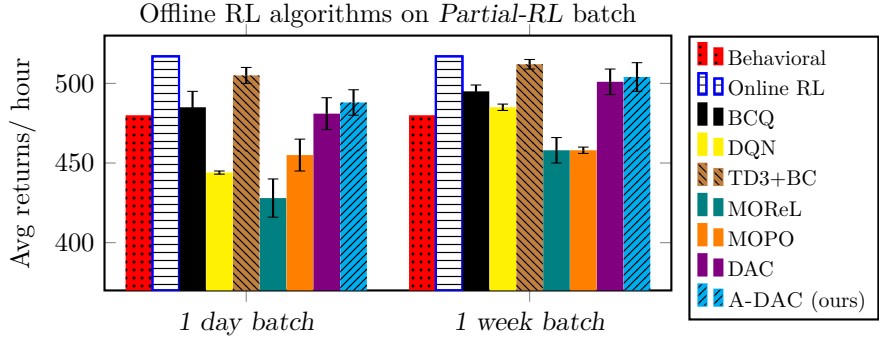

Figure 9: Comparison of offline RL algorithms on *Partial-RL* batch. Error bars indicate the min-max interval obtained after 5 runs with different seeds under the best model settings. Failures of DQN, MOReL, and MOPO exemplify the deadly triad issue in RL. Both DAC and A-DAC, on account of a more principled approximation of the dynamics manage to improve the behavior policy, the magnitude being higher on the larger batch. TD3+BC provides the best improvement where data from the partially trained policy aids with early identification of good actions on account of BC regularization.

of light traffic, one hour of medium traffic, and two hours of peak traffic. Each single hour episode measures the cumulative discounted rewards. The average return across the five hours is reported as our performance measure.

MDP derivation in DAC requires a nearest neighbor index: We use a memory-mapped fast approximate nearest neighbor search index (ann). Distances are max-normalized by diameter of the state space before deriving MDP. We use a sampling-based fast approximate algorithm to estimate the diameter. Derived MDP is solved optimally with a value iteration algorithm (mdp) which can exploit sparseness in transition matrix for efficiency gains.

**Baselines.** We compare A-DAC against recent approaches to offline learning. Firstly, we use three model-free offline RL baselines, namely, DQN, BCQ, and TD3+BC. Next, we use MOReL and MOPO as two representatives of the general MBPO (Janner et al., 2019) approach that covers both classical (naïve) MBRL and Pessimistic MDP-based MBRL. We first tune the parameters controlling pessimistic costs in each of

the algorithms and report the results for the best setting.[2] Finally, we analyze the DAC-MDP framework which, too, is tuned for cost parameter $C$ using a grid search as suggested in Shrestha et al. (2020). Some of the baselines we use here are originally developed for continuous action spaces; We describe the necessary modifications for our discrete action setup in Appendix C. An online RL policy is also included in the evaluation which uses an off-policy DQN fully trained for close to $100k$ timesteps.

## 6.3 Cyclic Policy Batch

Figure 8 compares the algorithms on the *Cyclic* dataset. Approaches that employ pessimism, *viz.*, BCQ and both MBRL algorithms, fail to improve the behavioral policy as the *Cyclic* policy does not offer much insight that the deep function approximators (employed either to learn the policy or the Q-values) can easily exploit. DAC-based policies provide considerable performance improvements due to the nearest neighbor-based dynamics approximation specialized for finite spaces like ours. For DAC, we only report results from the best hyper-parameter setting found after 6 online evaluations. As seen in Figure 1, the performance is highly sensitive to these settings. A-DAC, on the other hand, does not need extra online evaluations and either matches or equals the performance of the best DAC policy.

## 6.4 Partial RL Policy Batch

Figure 9 compares the offline RL algorithms on the *Partial-RL* dataset. BCQ manages to exploit this batch better with its constrained Q-value approximation. Surprisingly, the pessimistic MBRL approaches fail to even match the behavioral policy performance. Along with DQN, these suffer from the issue of "the deadly triad of deep RL" (Sutton & Barto, 2018) which exemplifies the inherent difficulties in planning with learned deep-network models. The incorporation of behavioral cloning into TD3 policy updates in TD3+BC makes the good actions stand out early during the training and, as an effect, it works much better here than on *Cyclic* data set. DAC framework simplifies the dynamics derivation process thus significantly reducing the reliance on the learned models. Between DAC and A-DAC, A-DAC, once again, matches or improves the best DAC policy performance without any hyper-parameter tuning.

## 6.5 State Representations in A-DAC

Results for A-DAC (and for DAC) presented so far, did not discuss the latent state representations used by the model. We analyze the following three state representations:
**1. Loop Counts**: 68 dimensional native representation, corresponding to counts from the loop detector devices.)
**2. BCQ**: Deep representation learned by BCQ algorithm. Penultimate layer of a learned Q-network is used as a surrogate high dimensional state.
**3. DQN**: Deep representation learned by DQN algorithm. The original state vector is mapped to a high dimensional space in a manner similar to BCQ.

Table 3 shows that the native state representation can exploit the simplistic *Cyclic* batch really well. For a *Partial-RL* batch, however, a representation learned by a strong offline exploration algorithm, such as BCQ, offers more benefits provided a sufficiently large sized batch is available. DQN uses a highly discriminative model which leads to a largely unchanged state representation with increasing data size. Results point at the flexibility of DAC framework to use state representations learned from other batch learning approaches and improve them further.

## 6.6 Efficiency and Hyperparameter Sensitivity

DAC results presented so far are obtained from the best policy from 6 policies trained with different hyperparameters based on a guideline given in Shrestha et al. (2020). It should be recalled that DAC requires two hyperparameters: a smoothness parameter $k$, and a cost penalty $C$. A-DAC uses the parameter $k$ as

---

[2]We use grid search to tune the parameters in both MOReL and MOPO that relate to behavior ranging from classical MBRL to highly pessimistic MBRL. It has been observed that depending on the nature of the data set (stationary or moving) or its size, the best settings differ. We only report the results obtained on the best tuned version here.

Table 3: Comparison of the state representations used in A-DAC. Error bars indicate the min-max interval obtained after 5 runs with different seeds. The *Cyclic* policy is improved significantly using native state representation. The *Partial-RL* batch is better optimized using the learned state representations.

| Batch type | Batch size | A-DAC state representation | | |
|---|---|---|---|---|
| | | Loop Counts | BCQ | DQN |
| *Cyclic* | *1 day* | **495 ± 5** | 473 ± 10 | 459 ± 12 |
| | *1 week* | 487 ± 7 | **507 ± 5** | 460 ± 10 |
| *Partial-RL* | *1 day* | 477 ± 8 | 460 ± 10 | **488 ± 8** |
| | *1 week* | 481 ± 5 | **504 ± 9** | 491 ± 4 |

Table 4: Comparison of offline RL algorithms based on the overhead required to build and evaluate policy given a *Cyclic* experience batch. DAC and A-DAC use the state representation from BCQ. An Online RL algorithm is also included for completeness.

| Algorithm | Time (minutes) | Number of timesteps |
|---|---|---|
| Online RL | 660 | 100k |
| BCQ | 99 | 24k |
| DQN | 101 | 25k |
| TD3+BC | 100 | 20k |
| MBRL | 65 | 18k |
| DAC | 180 | 28k |
| A-DAC | 113 | 24k |

well as a distance threshold $\alpha$ for smoothness while the cost penalty is not required. We compare sensitivity of the algorithms to each parameter in Appendix D. The main takeaways are presented next.

DAC has been shown to be robust to the smoothness parameter $k$; It holds for A-DAC's smoothness parameters too. We set a high value of $\alpha$, such as 0.8, and use a low value of $k$, 5 by default, which enables efficient computation while achieving robust results. DAC is highly sensitive to parameter $C$. A significant time overhead is required for online evaluations in order to tune this parameter. A-DAC offers robustness guarantees that are experimentally verified.

We compare the total time overhead for our baselines in Table 4 using conservative settings for evaluation stopping criteria or for the amount of policy optimization. In addition to the policy evaluation, we have to use significant computation overhead for building the MDP and solving it optimally. (See Appendix D for details.) With DAC, this overhead multiplies by the number of MDPs it derives in order to explore the best policy. While computational advances in future can potentially reduce this overhead, it is unlikely to be a non-trivial component of the overall optimization time.

## 7 Related work

### 7.1 Traffic Signal Control

Classical transportation engineering techniques for Traffic Signal Control (TSC) can be categorized as optimal control (D'ans & Gazis, 1976) or distributed network-of-queues control (Varaiya, 2013). The widely adopted adaptive traffic control systems such as Scats (Sims & Dobinson, 1980) largely depend on pre-defined rules or expert knowledge on traffic intersection management. There has been a lot of research work in recent years employing RL techniques to support dynamic traffic signal control (Rizzo et al., 2019; Wei et al., 2018; Nishi et al., 2018). These approaches develop different models for state and reward functions in order to optimize various objectives such as throughput, waiting time, or carbon emissions (Yau et al., 2017; Wei et al., 2019). Recent deep learning RL approaches such as deep policy gradient or graph convolutions have found success on more complex problems such as coordinated multi-agent control (Chen et al., 2020).

The related domain of Autonomous Vehicle (AV) control has recently seen a widespread use of RL for automation (Toghi et al., 2021; Wu et al., 2021), with some approaches even deploying the solutions at a small scale (Stern et al., 2018; Lichtlé et al., 2022). Shi et al. (2021) is the only work, to the best of our knowledge, that uses offline RL for AV control. Our work should be similarly treated as an important first step towards exploiting the big potential of offline RL in TSC.

The RL-based approaches for both TSC and AV control prominently use traffic micro-simulators. Many of the research projects use simulators built using multiple simplifying assumptions both on the road networks and the traffic flow models for efficiency reasons (Zhang et al., 2019; Fu et al., 2019). We make use of a mature open-source micro-simulation framework of Sumo (Lopez et al., 2018) which allows us to simulate real road network with traffic modeled on the previously collected statistics generated by loop detector devices installed in the real environment.

### 7.2 Offline Model-free RL

Off-policy model-free RL methods (Mnih et al., 2016) adapted to batch settings have been shown to fail in practice due to extrapolation errors (Fujimoto et al., 2019b). Most of the offline RL algorithms are built on top of an existing off-policy deep RL algorithm, such as TD3 (Fujimoto et al., 2018) or SAC (Haarnoja et al., 2018). In order to handle out-of-distribution errors, they apply various regularization measures such as actor pre-training with imitation learning (Kumar et al., 2020), generative behavior cloning models (Wu et al., 2019; Kostrikov et al., 2021a) KL-divergence (Kumar et al., 2019; Jaques et al., 2019), ensemble uncertainty quantifiers (Agarwal et al., 2020), behavior support constraints (Kostrikov et al., 2021b; Fujimoto et al., 2019b) or behavior cloning (Fujimoto & Gu, 2021). Each of these algorithmic modifications for conservatism add supplementary hyper-parameters which may need to be tuned. In offline settings, however, it is not feasible to interact with the environment for tuning. Our approach has a built-in adaptive pessimism which empowers it to work out-of-the-box.

### 7.3 Offline Model-based RL

Compared to model-free, the model-based offline approaches have proven to be more data-efficient while also benefiting from more supervision (Levine et al., 2020; Yu et al., 2020). Our work builds on this recent success of MBRL to offline batches. Uncertainty quantifiers are critical for generalization of the model (Argenson & Dulac-Arnold, 2020). Reward penalties act as a strong regularizer (Yu et al., 2020; Kidambi et al., 2020) and fits naturally to the nearest neighbor approximation used in our MDP model (Shrestha et al., 2020). The strong approximation of the dynamics and optimal planning on account of finite problem structure are key to our approach avoiding the deadly triad of deep RL (Sutton & Barto, 2018). Further, our adaptive mechanism for shaping the reward penalties enable a high quality out-of-the-box performance while the other approaches are sensitive to hyper-parameters and require a careful parameter tuning Lu et al. (2021).

## 8 Conclusion

We have modeled traffic signal control as an offline RL problem and learnt a policy from a static batch of data without interacting with a real or simulated environment. The offline RL is a more realistic set up as it is practically infeasible to learn a policy by interacting with a real environment. Similarly in a simulator it is not clear how to integrate real data that is often available through traffic signal operators.

We have introduced a model-based learning framework, A-DAC, that uses the principle of *pessimism under uncertainty* to adaptively modify or shape the reward function to infer a Markov Decision Process (MDP). Due to the adaptive nature of the reward function, A-DAC works out of the box while the nearest competitor requires substantial hyperparameter tuning to achieve comparable performance. An evaluation is carried out on a complex signalized roundabout showing a significant potential to build high performance policies in a data efficient manner using simplistic *cyclic* batch collection policies. In future, we would like to explore other applications in the traffic domain which can benefit from offline learning.

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

# A  Proof for Optimality Guarantee

In this section, we present the proof for Theorem 4.3 using the construction proposed in Pazis & Parr (2013) for Probably Approximately Correct (PAC) exploration in continuous state MDPs, with some additional modification due to our pessimistic setting.

We assume that the rewards lie in $[0, R_{max}]$. We further assume local Lipschitz continuity of the $Q$-function as defined in Assumption 4.2. The local Lipschitz continuity assumption allows us to use samples from nearby state-actions as required by A-DAC model and approximate Q-function without much error.

**Definition A.1.** For a state-action pair $(s_i, a_i)$, the pessimistic $Q$-value function used in A-DAC MDP $\tilde{M}$ is defined as:

$$\tilde{Q}(s_i, a_i) = \frac{1}{k} \sum_{j \in NN(s_i, a_i, k, \alpha)} \left( \max\{0, r_j + \gamma \, \tilde{V}(s'_j) - r_{max} d'_{ij} \} \right) \tag{7}$$

where $d'_{ij} := d(s_i, a_i, s_j, a_j)$ max-normalized by the diameter of the state-action sample space, $\alpha$ is the distance threshold used in the nearest neighbor function, and $r_{max} = \max_{j \in NN(s_i, a_i, k, \alpha)} r_j$.

The number of samples required to get a good approximation depends on the covering number of the state-action space $\mathcal{N}_{\mathcal{SA}}(\alpha)$ defined next.

**Definition A.2.** The covering number $\mathcal{N}_{\mathcal{SA}}(\alpha)$ of a state-action space is the size of the largest minimal set $C$ of state-action pairs such that for any $(s_i, a_i)$ reachable from the starting state, there exists $(s_j, a_j) \in C$ such that $d'_{ij} \leq \alpha$.

Let $\tilde{\pi}$ be the optimal policy of A-DAC MDP $\tilde{M}$. Our goal is to bound the value $V^{\tilde{\pi}}(s)$ of $\tilde{\pi}$ in the true MDP $M$ in terms of the optimal value $V^*(s)$ for any state $s$. The following lemma suggests that it is sufficient to bound the Bellman error $\tilde{Q}(s, a) - B[\tilde{Q}](s, a)$ across all $(s, a)$ pairs with respect to the true MDP.

**Lemma A.3.** *[Theorem 3.12 from Pazis & Parr (2013)] Let $\epsilon_- \geq 0$ and $\epsilon_+ \geq 0$ be constants such that $\forall (s, a) \in (\mathcal{S}, \mathcal{A}), -\epsilon_- \leq Q(s, a) - B[Q](s, a) \leq \epsilon_+$. Any greedy policy $\pi$ over a Q-function $Q$ then satisfies:*

$$\forall s \in \mathcal{S}, V^\pi(s) \geq V^*(s) - \frac{\epsilon_- + \epsilon_+}{1 - \gamma}$$

In order to use this lemma, we want to bound the quantity $Q(s, a) - B[Q](s, a)$ for a fixed point solution $\tilde{Q}$ to the pessimistic Q-function in Definition A.1[3]. For a locally Lipschitz continuous value function (Assumption 4.2), the value of a state-action pair can be expressed in terms of any other state-action pair in its neighborhood as $Q(s_i, a_i) = Q(s_j, a_j) + \xi_{ij} L_Q(i, \alpha) d'_{ij}$, where $\xi_{ij}$ is a fixed but possibly unknown constant in $[-1, 1]$. For a sample $(s_j, a_j, r_j, s'_j)$, let

$$x_{ij} = r_j + \gamma V(s'_j) + \xi_{ij} L_Q(i, \alpha) d'_{ij}$$

Then:

$$E_{s'_j}[x_{ij}] = E_{s'_j}[r_j + \gamma V(s'_j)] + \xi_{ij} L_Q(i, \alpha) d'_{ij}$$
$$= Q(s_j, a_j) + \xi_{ij} L_Q(i, \alpha) d'_{ij}$$

Consider a state-action pair $(s_0, a_0)$ and its $k$-nearest neighbors given by $NN(s_0, a_0, k, \alpha)$, we can estimate the $Q$-value for the pair by averaging over the predicted values of its neighbors:

$$\hat{Q}(s_0, a_0) = \frac{1}{k} \sum_{i \in NN(s_0, a_0, k, \alpha)} x_{0i} \tag{8}$$

Let us define a new Bellman operator $\hat{B}$ corresponding to the definition of $\hat{Q}$ above. While $B$ denotes the exact Bellman operator, $\tilde{B}$ denotes the approximate Bellman operator for the pessimistic value function in Definition A.1.

---

[3]It can be easily proven that $\tilde{Q}$ has a unique fixed point by showing that the Bellman operator $\tilde{B}$ is a contraction in maximum norm.

The Bellman error can be decomposed into two parts: (a) the maximum sampling error $\epsilon_s$ caused by using a finite number of neighbors, and (b) the estimation error $\epsilon_d$ due to using neighbors at non-zero distance. The following lemma from Pazis & Parr (2013) bounds the minimum number of neighbors $k$ required to guarantee certain $\epsilon_s$ with probability $1 - \delta$:

**Lemma A.4.** *[Lemma 3.13 from Pazis & Parr (2013)] If* $\frac{\tilde{Q}_{max}^2}{\epsilon_s^2} ln\left(\frac{2\mathcal{N}_{\mathcal{SA}}(\alpha)}{\delta}\right) \leq k \leq \frac{2\mathcal{N}_{\mathcal{SA}}(\alpha)}{\delta}$,

$$\forall(s,a), |\hat{B}[\tilde{Q}](s,a) - B[\tilde{Q}](s,a)| \leq \epsilon_s, \quad w.p. \quad 1 - \delta$$

The proof (not included here) applies Hoeffding's inequality to bound the difference in the true expectation (given by operator $B$) and its estimation using mean over $k$ samples (given by operator $\hat{B}$).

The second piece of the Bellman error requires us to bound the term $\epsilon_d = \tilde{B}[\tilde{Q}](s,a) - \hat{B}[\tilde{Q}](s,a)$.

**Lemma A.5.** *For all known state-action pairs* $(s,a)$

$$0 \leq \tilde{B}[\tilde{Q}](s,a) - \hat{B}[\tilde{Q}](s,a) \leq \bar{d}_{max}R_{max}$$

*Proof.* We can simplify the estimation error $\epsilon_d$ by using the definitions (7) and (8).

$$\epsilon_d = \tilde{B}[\tilde{Q}](s_i, a_i) - \hat{B}[\tilde{Q}](s_i, a_i)$$
$$= \frac{1}{k} \sum_{j \in NN(s_i, a_i, k, \alpha)} \left( -r_{max} - \xi_{ij}L_Q(i,\alpha)\right) * d'_{ij}$$

We can set $\xi_{ij} = -\frac{R_{max}}{L_Q(i,\alpha)}$ which will bound the quantity inside the bracket from $[0, R_{max}], \forall(s,a)$ since $0 \leq r_{max} \leq R_{max}$. The worse case average distance is defined as $\bar{d}_{max}$, therefore ensuring that $\epsilon_d \leq \bar{d}_{max}R_{max}$. $\qquad \square$

Finally, to prove the PAC bound in Theorem 4.3, we need to bound the quantity $\tilde{Q}(s,a) - B[\tilde{Q}](s,a)$. To achieve that, we combine the above two lemmas and apply the operators on the fixed point solution $\tilde{Q}$, giving us :

$$\text{If } \frac{\tilde{Q}_{max}^2}{\epsilon_s^2}ln\left(\frac{2\mathcal{N}_{\mathcal{SA}}(\alpha)}{\delta}\right) \leq k \leq \frac{2\mathcal{N}_{\mathcal{SA}}(\alpha)}{\delta}, \forall(s,a), -\epsilon_s \leq \tilde{Q}(s,a) - B[\tilde{Q}](s,a) \leq \epsilon_s + \bar{d}_{max}R_{max}$$

Putting these bounds in Lemma A.3 gives us the final result. $\qquad \square$

## B    Modeling Environment for a Signalized Roundabout

We model a signalized roundabout (Figure 7) used previously to learn an online RL policy using policy gradient Rizzo et al. (2019). It consists of three types of lanes or traffic arms: (a) approaching/ incoming lanes, (b) outgoing lanes, and (c) circulatory lanes that enable traffic flow redirection. Each traffic arm has multiple lanes. It is important to consider each lane separately because the way the incoming traffic 'weaves in' to the circulatory lanes impacts the wait time of vehicles. Movement in or out of the circulatory lanes is controlled by traffic signals numbering 10 in total.

**States.** A state corresponds to the number of vehicles counted by the loop detector devices installed on every lane. For each approaching or outgoing lanes, there are two devices per lane, one close to the roundabout and another several meters farther. The detectors number 68 in total.

**Actions.** Actions correspond to traffic control phases activated for a fixed time duration. A phase is provided for each set of non-conflicting flows. For example, traffic moving from north to south and from south to north does not conflict and therefore constitutes a single phase. In all, we use a discrete set of 11 actions as modeled in Rizzo et al. (2019).

**Rewards.** Traffic signal control typically serves a dual objective: maximize throughput and avoid long traffic queues. We optimize only the first objective here and leave the the later for a future demonstration. For throughput maximization, the rewards are modeled as the cumulative capacity: The cumulative capacity $C(t)$ at time $t$ is the number of vehicles that left the roundabout from time 0 to $t$. Reward at time step $t$ is then defined as $R(t) = C(t) - C(t-1)$.

**O-D Driven Traffic Simulation:**  Access to real experience trajectories data is often very limited and/ or allows only a specific behavioral policy (e.g. *Cyclic* in our case) offering little room for experimentation. We describe the process employed to augment the batch collection process in our environment.

A micro-simulator is set up using real network configuration with traffic signals and loop detector devices correctly placed. Traffic is generated using an Origin-Destination (O-D) matrix Peterson (2007) which is prepared by urban planning authorities. The O-D matrix corresponds to macro statistics for a relatively small, but significantly larger than the traffic phase duration, period of the day. It enumerates the number of vehicles that move between each pair of traffic zones positioned in the close vicinity of the roundabout. The data is collected by routine monitoring of traffic by loop detector devices and is believed to be stable. The O-D data is fed to Sumo micro-simulator which generates vehicle routes following the provided source, destination, and frequency requirements. Such a real demand-driven simulation is used for batch data augmentation.

## C  Evaluation Baselines

A-DAC is evaluated against three RL approaches that are specifically created for discrete action spaces, namely, (a) DQN (Mnih et al., 2015), (b) BCQ (Fujimoto et al., 2019b) which has both a continuous action and a discrete action version, and (c) DAC-MDP (Shrestha et al., 2020). We use the author provided implementations for each of these. In addition to these baselines, we also include some recent approaches built specifically for continuous action spaces. We briefly describe the modifications needed in these algorithms to make them work for discrete action spaces.

### C.1  Discrete TD3+BC

TD3 (Fujimoto et al., 2018), short for Twin Delayed DDPG, is a deep off-policy RL algorithm that can only be used in continuous action environments. It provides three important features that make it a strong RL baseline: (a) Clipped double-Q learning, (b) Delayed policy updates, and (c) Target policy smoothing. For offline algorithms, a behavior cloning regularization is applied (Fujimoto & Gu, 2021) to TD3 policy training as shown in Eq. 4. We make the following modifications to the TD3+BC algorithm:

- The two Q-functions and the policy function take the form $Q_{\phi_i} : \mathcal{S} \to \mathbb{R}^{|\mathcal{A}|}, i \in \{1, 2\}$ and $\pi_\theta : \mathcal{S} \to [0, 1]^{|\mathcal{A}|}$.

- The loss function for policy network $\pi_\theta$ is given by $L(\theta, \mathcal{D}) = \mathbb{E}_{(s,a)\sim\mathcal{D}}\, \pi_\theta(s)^T\left(-\lambda Q_{\phi_1}(s)\right) + \left(\pi_\theta(s) - a\right)^2$, where $\lambda$ is a normalizing scalar.

- Smoothing of action selection from policy network for target $Q$ evaluation is disabled because it is possible to calculate the exact action distribution in discrete setting.

- Each of the critic (Q) networks is trained with the loss function given by $L(\phi_i, \mathcal{D}) = \mathbb{E}_{(s,a,s',r)\in\mathcal{D}}[\left(Q_{\phi_i}(s) - y(r, s')\right)^2]$, $i \in \{1, 2\}$ where the target $y$ is given by $y(r, s') = r + \gamma min_i\{\pi_\theta(s')^T Q_{\phi_i}(s')\}$

Please see Christodoulou (2019) for a similar exercise on deploying the equally popular SAC (Haarnoja et al., 2018) algorithm on discrete action spaces.

### C.2  Discrete MBPO

We evaluate two algorithms that first build an approximate MDP dynamics from the offline data set and then do sample rollouts from the derived MDP to optimize a policy network. The first, MOReL (Kidambi

et al., 2020), derives an ensemble of learned dynamics models Nix & Weigend (1994) which allows tracking of uncertainty in estimation of next state. This uncertainty quantification is used in creating a pessimistic(P-) MDP model where transitions to *uncertain* regions are restricted by a threshold parameter. While this P-MDP can be solved using any planning algorithm, authors train a policy using model-based natural policy gradient (Rajeswaran et al., 2020). Given this infrastructure, only minimal changes are required to run this algorithm on discrete action setup. The dynamics models use a single dimension for action input. Further, the policy network predict a probability distribution over all possible actions.

The second, MOPO (Yu et al., 2020), starts by building an ensemble of learned dynamics models similar to MOReL. Further, it learns a reward model from the data set as well. It then derives a new uncertainty-penalized MDP: the uncertainty quantification given by the ensemble of models is used to penalize reward estimates. This is followed by policy optimization using model rollouts. Similar to MOReL, we use a single dimension action input for dynamics model and then use a DQN network to learn an optimal policy for the derived MDP.

## D   Additional Evaluation

### D.1   Hyperparameter Sensitivity of DAC

We analyze the sensitivity of each hyperparameter individually. A-DAC retains DAC's robustness to smoothness parameter $k$; Figure 10 provides the evidence. For a small value of $k$ (ranging between $2 - 10$), we find that parameter $\alpha$ has a minimal role. For instance, Figure 11 studies the impact of $\alpha$ when $k$ is set to 5. Except for very low values of $\alpha$, the performance remains unaffected. It should be noted that having a large distance threshold does not harm since our adaptive reward computation penalizes distant neighbors. Based on this, we set $\alpha$ to 0.8 by default.

When it comes to parameter $C$, Figure 12 shows that DAC is highly sensitive to the parameter. The values for $C$ are varied between the minimum and the maximum rewards observed in the dataset. The robustness offered by A-DAC by adapting rewards based on local neighborhood ensures that A-DAC does not need to spend expensive cycles on hyperparameter tuning.

### D.2   Computational Overhead

A breakdown of computation time overhead is presented in Figure 13. All numbers are obtained from a server running a 16-core 2nd generation Intel Xeon processor with 128GB RAM. Our implementation uses fast approximate solutions to nearest neighbor searches and diameter computations. But the MDP build and MDP solve operations suffer as they process a transition matrix growing quadratically with the number of core states. Designing a distributed GPU based implementation for optimal planning is left as a future work.

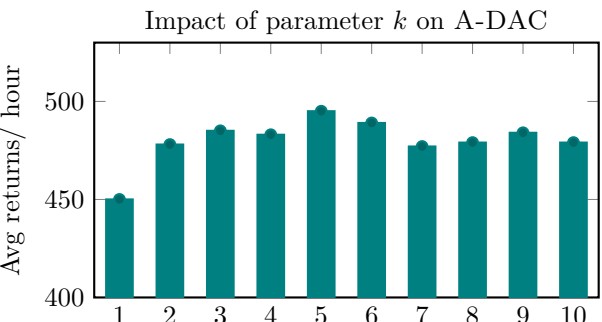

Figure 10: Analyzing the impact of smoothness parameter $k$ in A-DAC. The setting $k = 1$ makes the MDP deterministic and incapable of exploiting the different transitions observed in the experience data. Performance is not much sensitive to values of $k > 1$.

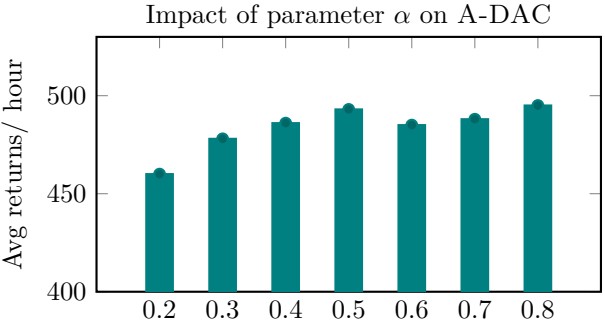

Figure 11: Analyzing the impact of smoothness parameter $\alpha$ in A-DAC. Low settings result in insufficient neighbors used in approximation that has an adverse effect. High settings are more robust.

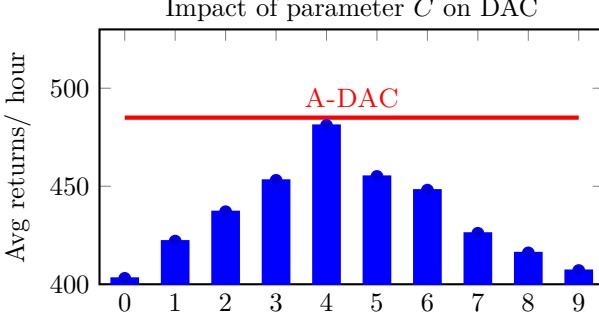

Figure 12: Comparing the impact of cost penalty $C$ in DAC to the adaptive reward penalties in A-DAC. DAC is highly sensitive to $C$ and requires a careful tuning. A-DAC manages to match or better the performance of the best $C$ setting in DAC out-of-the-box.

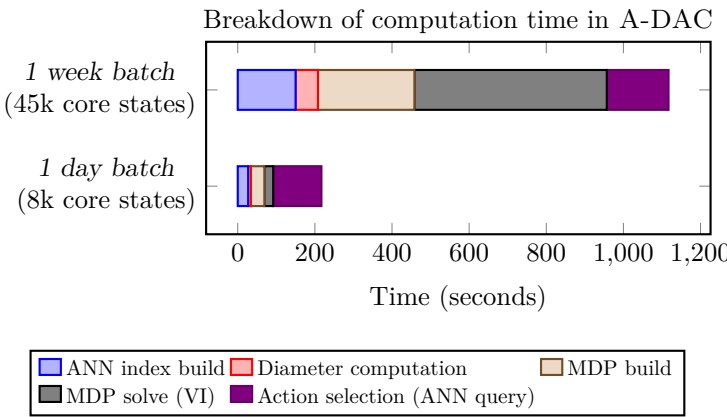

Figure 13: Computation time breakdown between different processes in A-DAC. The nearest neighbor querying and the diameter computation use fast approximate algorithms and scale well. But MDP build and solve stages suffer from quadratic time complexity.

