# OpenReview forum: "Offline Reinforcement Learning for Traffic Signal Control"
_TMLR — Rejected by TMLR_

### Review · Reviewer_da7k · 2022-07-17

**Summary Of Contributions:**

The paper makes several contributions:
1. They provide a variant of DAC they call A-DAC.
2. They propose using offline RL for the traffic light benchmark.
3. They show how to collect real world data from macro statistics.



**Broader Impact Concerns:**

Not relevant.

**Requested Changes:**

Critical:
DAC vs. A-DAC clarification:
- $d'(s, a, s_i ,a_i)$ is not properly defined (it is the max-normalized version of d, but not sure what that means). Why is it used instead of d?
- How do Theorem 4.3 compares to DAC results?
- Empirically, the evaluation results of DAC and A-DAC look very similar, why is A-DAC faster?
- DAC requires parameter tuning for C, but A-DAC doesn't. Is this the main selling point?

Do you intend to release an offline RL benchmark?
How well does it allow comparing different algorithms, and how close to real world is it?

Strengthen:
Can you propose adaptation or just relate to the non-stationarity usually happening in real world?



**Strengths And Weaknesses:**

Strengths
The paper is written pretty clearly, it provide an important application which sits well with RL. Many types of RL algorithms are compared.

Weaknesses
The authors invest a great deal in their A-DAC MDP formulation. However, I'm afraid it is not very clear to me that it is better than DAC, or when.
- $d'(s, a, s_i ,a_i)$ is not properly defined (it is the max-normalized version of d, but not sure what that means). Why is it used instead of d?
- How do Theorem 4.3 compares to DAC results?
- Empirically, the evaluation results of DAC and A-DAC look very similar, why is A-DAC faster?
- DAC requires parameter tuning for C, but A-DAC doesn't. Is this the main selling point?

Also, for contribution 2 (offline RL for traffic light benchmark), the authors should release a code with data to solicit other researchers to use the benchmark, I'm not sure whether it is done but didn't see it mentioned.

I wonder if there was no previously use benchmark that could have been compared on (even if it was used primarily for RL) just to make sure the problem is interesting.

I also think an important factor of traffic control relates to multiple traffic signals and their effect on one another, as well as non-stationarity which coincides with the offline assumptions. I wonder if these topics should have been discussed or considered for this specific setup.

---

> ### Author Response · Authors · 2022-07-22
> **Response to Review of Paper253 by Reviewer da7k**
>
> Thank you very much for the insightful comments. We are happy to clarify the issues raised above and make necessary corrections. Please find our response below:
>
> 1. Distance needs to be normalized to bring the penalty term in the units of rewards. The Theory is not tied to any specific normalization strategy, but for practicality we use max-normalization ($d’ = d / d_{max}$) . In particular, we approximately calculate ‘diameter’ ($d_{max}$) of the euclidean space formed by all points by sampling a few points and comparing distance from these points to all points in space. We are adding a note under Definition 4.1 describing this process.
>
> 2. Theorem 4.3 largely uses a similar theoretical foundation as DAC; in particular, both use Theorem 3.12 from [Pazis & Parr (2013)] to bound the values given by $V^\pi$ (See Appendix A). A-DAC uses two different assumptions in its derivation though: (a) Local Lipschitz continuity is assumed on $Q$-function which is less restrictive than global Lipshitz continuity, and (b) Nearest neighbor queries are enforced within a distance threshold \alpha which means we consider only $up\ to\ k$ neighbors rather than $exactly\ k$ neighbors used in DAC. The implication of the second assumption is that we are able to use the ‘covering number’ of the state-action space (Definition A.2) as a measure of volume of the data. This simplifies the intuitive understanding of the tightness bounds where the first factor relates to the smoothness factor $k$ while the second factor relates to the size of the dataset $D$. However, it’s hard to establish between the DAC and A-DAC bounds which one is tighter as they are expressed in different terms though having an $r_{max}$ as opposed to a hyper-parameter tuned $C$ in the bound is  more meaningful.
>
> 3. A-DAC is faster precisely because it doesn’t need to train and evaluate multiple models with different hyper-parameters; the choice of $r_{max} \times d’$ as the penalty (Definition 4.1) proves to be a robust choice that gives us at least as much performance as the one by a well-tuned DAC. The performance graphs shown in Figure 8 and Figure 9 compare A-DAC to the best performing DAC and show that A-DAC can beat a well-tuned DAC. It should be considered here that these improvements are obtained using a significantly smaller time overhead as can be seen in Table 4. We will make this point clear in Section 6.3.
>
> 4. A-DAC code and data is attached along with the submission. Here’s an anonymous github link: https://anonymous.4open.science/r/ADAC-traffic-4C1E/README.md. We are open sourcing our data and our simulation environment to be used as a benchmark. Scripts to reproduce results on prior baselines will be packaged soon.
>
> 5. We have built the infrastructure to model the simulation based on non-stationarity of the real world. It uses statistics obtained from traffic authorities to simulate traffic observed by the complex intersection shown in Figure 7. The process is outlined in Appendix B. Further, our evaluation picks 5 different hours during the day and plots the average returns to test the robustness of the algorithm to varying traffic.
>
> 6. There are two problems with previous traffic control benchmarks: (a) They do not model traffic based on real data or; (b) They model simple road networks (see Section 2 for an example). There is, however, a strong line of work on multi-agent online RL for city-wide networks (e.g., [Chen et.al. Toward A Thousand Lights, AAAI’20], and [Ault & Sharon, RESCO, NeurIPS'21]) which we want to explore using offline RL in an independent work.

---

> > ### Comment · Reviewer_da7k · 2022-08-01
> > **Thank you for your response**
> >
> > I appreciate you releasing the code the data and answering my questions.

---

### Review · Reviewer_fEex · 2022-07-26

**Summary Of Contributions:**

This paper considers the problem of offline RL for traffic light control, an important problem with potential high impact. The work seems useful given that it is using recent methods on a relevant problem, thus bridging the gap between SoTA offline RL and applications. At present I do not think the paper can be accepted simply because it makes such a feeble attempt to cite relevant literature. The paper draws on ideas from a variety of areas and yet only cites 25 previous works in total. This seems negative for the field/science as readers new to the area would be left 1) not knowing the full academic ancestry of the work 2) thinking this idea was more novel than it is. There are also some issues regarding the baselines being under tuned. I would prefer to see the paper presented as "Offline MBRL can work for traffic light control" rather than "We propose a new *method* for offline RL and apply it to traffic light control".

**Broader Impact Concerns:**

It would actually be useful to explain further how this could have broader impact. Right now it is just another offline RL from sim-to-data-to-sim. We have many of these already! Could this actually be deployed? If so, then what are the implications?

**Requested Changes:**

1) The biggest issue in my mind is the lack of citations for other, related works. This is the case throughout the paper, both in terms of citing fundamental ideas as well as other papers with relevant recent ideas. The goal of TMLR is to remove focus on novelty and prioritize good science, so citing these exclusively adds to the contribution in my opinion. Below are the main ones I can think of:
- Add a cite for things in the preliminaries like Bellman backup error. The first part of section 3 cites a total of one paper, and it introduces many fundamental concepts. Just because methods are established does not mean they should not be given appropriate recognition.
- There is no citation for TD3 when it is first mentioned.
- There is no citation for MBPO.
- There is no citation for Dyna/MBRL in general.
- There is no citation for [1] (above) which proposes an adaptive penalty for offline MBRL.

2) The model free baselines seem old. There is no CQL, Fisher BRC or IQL. I would expect these to outperform even further on the partial RL dataset.

3) I would hope to see a tuned version of MOPO or for example, MOPO + the same adaptive conservatism bonus. Otherwise it is unclear why we are using this DAC approach.

**Strengths And Weaknesses:**

Strengths:
- The promise of the paper is very exciting, I am just not convinced it delivers (see Weaknesses).
- I am not a theoretician and do not place tremendous value in theoretical results that are included in primarily empirical papers. However, in this case it is certainly useful because we may be deploying these policies in the real world. I also appreciate the local smoothness assumption (vs. global) may be reasonable for the traffic example. For the next version it could also be worth commenting on the property that the bound gets tighter when the dataset includes more *diverse* data, since $\overline{d}_\text{max}$ will presumably only decrease if the increase in data is distinct from the existing dataset. This is somewhat obvious but may make the read a little clearer.
- It is very helpful to have a toy example feature prominently in the paper.

Weaknesses:
- The adaptive conservatism has been proposed before in [1]. The issue here is not novelty, because we do not require that for TMLR, it is just bad science to claim to be the first to do something when the exact idea was presented this year at a conference.
- The nearest neighbor-based dynamics approximation is incredibly specific and likely only works for low dimensional discrete problems. Indeed, it seems sensitive to the representation used. I gather the baselines did not have the benefit of selecting the max over three different representations!
- How were the baseline hyperparameters tuned? It seems like DAC is very sensitive to its own hyperparameters (e.g. Figure 12) so it is possible the baselines simply were under tuned given they have not been used in this setting before. In addition, I would be interested to see a tuned version of MOPO instead of MOReL for comparison, since I am not convinced MOReL is superior, as they used a different version of D4RL. I don’t find it “surprising” that MOReL fails here at all since it has many moving parts and was likely overfit to the specific problems in the paper.
- The most exciting part of the project is the idea that we can learn simulators from offline data and then use them to train RL policies for tasks where we don’t have a simulator. If this was the first application of this idea it would be exciting. However, it seems the actual experiment conducted does indeed use a simulator to collect the data. So you have to ask, what is the point? Of course it could be the case that it also works for real data, but, as of yet we have no evidence that this is true. Given that the actual method is not new, this just seems like a paper applying a known method to a different simulated dataset.
- One of the main claims is that this is learning an MDP for a practical real world problem. There is no mention of the fact that recent works have actually deployed policies trained from offline data in *similar* settings, which is drastically more impressive than just showing it works in a simulator. For example [2].
- As mentioned in the “Requested Changes” section, it is a huge weakness to not cite a large volume of other works in related areas. This alone is grounds for rejection in my opinion as it is bad for the field.
- What does the reward shaping term learn? I would expect to see analysis of this given that it is the only change in the paper vs. the previous method (DAC).
- How does this scale to higher dimensional problems, e.g. with multiple similar discrete actions? What about continuous?

Minor issues
- In the third paragraph the authors say “online or off-policy” this is not a contrast. Off-policy methods can be online or offline.
- The citations are incorrect throughout, there should be parentheses around them when they are not part of the sentence, using \citep. For example: “The AlphaGo agent was very good at Go (Silver, 2016)”. Please correct this.
- Related to the above, it should be like this for abbreviations (RL, Sutton and Barto (2018))
- “The offline policy building is…” → “Offline policy building is…”
- Sec 3 first line no capitals for Reinforcement.
- In the paragraph below Eqn 2, “episodes/ trajectories” there shouldn’t be a space there.
- “called observation period.” → “called an observation period.”
- “MBRL primarily learn an” → “MBRL primarily learns an”
- “can exploit simplistic Cyclic batch” → “can exploit the simplistic Cyclic batch”

[1] Lu et al. Revisiting Design Choices in Model-Based Offline Reinforcement Learning. ICLR 2022

[2] Lichtlé et al. Traffic Smoothing Cruise Controllers Learned from Trajectory Data. ICRA 2022

---

> ### Author Response · Authors · 2022-08-11
> **Response to Review of Paper253 by Reviewer fEex (1)**
>
> > Strengths:
>
> Thanks for appreciating our work! We have incorporated your observation on diverse data at the end of Section 4.
>
> > The adaptive conservatism has been proposed before in [1]. The issue here is not novelty, because we do not require that for TMLR, it is just bad science to claim to be the first to do something when the exact idea was presented this year at a conference.
>
> The work in [1] is clearly not about adaptive conservatism as we have proposed in our paper. In [1], the main contribution is to use Bayesian Optimization for hyper-parameter tuning of the pessimistic coefficient and study its interaction with other hyperparameters in the system. In fact [1] can serve as an additional motivation for our work - i.e., to avoid addition of another hyperparameter to control uncertainty penalty (arguably the most important hyperparameter) in the set up. Our revised introduction reflects this additional motivation along with the recent comments made by Andrew Ng about the sensitivity of hyperparameter tuning on RL solutions.
>
> > The nearest neighbor-based dynamics approximation is incredibly specific and likely only works for low dimensional discrete problems. Indeed, it seems sensitive to the representation used. I gather the baselines did not have the benefit of selecting the max over three different representations!
>
> The nearest-neighbor distance function assumes a discrete action space but not a discrete state space. In our case the state-space is 68-dimensional (for 68 detectors). The actual data lives in a much lower-dimensional space, because of correlation between the detectors, making the nearest neighbor operation both statistically and computationally feasible.
>
> The question of which representation to use was not addressed in DAC and left for future work. We do not provide a full solution either, but through an empirical analysis presented in Table 3, a couple of guidelines are provided for the TSC use case: (a) Native (detector provided) state representation is aligned well with Cyclic data batch, and (b) State representation given by model-free offline algorithms such as BCQ offers more benefits when the batch is generated by partial-RL. We follow these guidelines in configuring the appropriate representation at the start of training.
>
> > How were the baseline hyperparameters tuned? ... it is possible the baselines simply were under tuned given they have not been used in this setting before. In addition, I would be interested to see a tuned version of MOPO instead of MOReL for comparison...
>
> General issue with tuning prior algorithms: There are possible implementation choices for the likes of network architecture, learning rates, or pre-training that impact the results. Both TD3+BC paper and Fisher-BRC paper point out the differences in results that they observe when repeating experiments from author-provided code and their own implementations for some of the baselines. We restrict our tuning of the baselines to only the pessimism/uncertainty parameters. Notes are added to that effect in Section 6.2 under para `Baselines’.
>
> We have added MOPO baseline now which is tuned for the hyperparameter $\lambda$. The results, however, are not very different than MOReL. (See Figure 8 and Figure 9.) We observe that the uncertainty estimation from the ensemble of deep models for dynamics used in both approaches tends to be highly inaccurate. A simpler, much controlled, uncertainty estimation through nearest neighbors, on the other hand, proves to be more accurate in our discrete action setting.
>
> > The most exciting part of the project is the idea that we can learn simulators from offline data ... it seems the actual experiment conducted does indeed use a simulator to collect the data. So you have to ask, what is the point? ...
>
> Our approach is not “sim-to-data-to-sim” because we start with a  “real” origin-destination(OD) matrix that was obtained from local traffic authorities. We use the SUMO simulator to generate routes from “real” data which are consumed by A-DAC to infer a policy. Offline RL has a better chance to be deployed because the current commercial ecosystem is dominated by traffic signal controllers based on classical control methods. Traffic authorities are wary of trying out  “trial and error” approaches to generate data, in real settings. We are continuing to engage with the local traffic authorities who provided us with the OD matrix data on this track.
>
> > One of the main claims is that this is learning an MDP for a practical real world problem. There is no mention of the fact that recent works have actually deployed..
>
> While RL for Automated Vehicle control approaches have seen a few test deployments as is correctly pointed out (Our related work section 7.1 is now updated with more examples like [2]) , RL for TSC approaches have faced challenges in deployment as mentioned in response to the earlier question. This could change in future though with application of offline RL methods.

---

> > ### Comment · Reviewer_fEex · 2022-08-16
> > **clarifications**
> >
> > There are some incorrect and questionable statements here so I am going to reply to these alone:
> >
> > >It should be clarified that the goal of our paper is primarily to solve a real world use case using offline RL
> >
> > This is not at all how you wrote the paper. If that is your goal then you need to re-write the paper and remove the sections about how ADAC is novel and theory for why it is a superior method. If instead, a handful of 2021 methods would do just as well, then why do we need ADAC? Either 1) remove ADAC and just test the recent sota baselines and present the work as offline RL in a new problem or 2) justify why we need ADAC. At the moment it is doing the worst of both.
> >
> > >The work in [1] is clearly not about adaptive conservatism as we have proposed in our paper. In [1], the main contribution is to use Bayesian Optimization for hyper-parameter tuning of the pessimistic coefficient and study its interaction with other hyperparameters in the system.
> >
> > I never said [1] was *about* adaptive conservatism, but *it is introduced and therefore not a novel idea* so you need to change the framing in your paper. I am fine with ideas being re-used, that is of course natural in science, but do not present ideas as novel when they are not. For clarity, look in [Section 6](https://arxiv.org/pdf/2110.04135.pdf)
> >
> > "We tune the penalty weight on-the-fly to a constraint value of Λ = 1, meaning we use only a single hyperparameter across all environments. Full details on the penalty weight tuning are provided in App. I"... I would encourage the authors look at App. I.
> >
> > Second, [1] is not about BO for hp tuning, the fact you interpret it that way is symptomatic of the community. The paper is clearly about understanding hyperparameter choices for offline model based RL, and then the BO part simply shows that if you do tune the hyperparameters you can outperform.

---

> > > ### Author Response · Authors · 2022-08-16
> > > **Response to clarifications**
> > >
> > > > This is not at all how you wrote the paper. If that is your goal then you need to re-write the paper and remove the sections about how ADAC is novel and theory for why it is a superior method. If instead, a handful of 2021 methods would do just as well, then why do we need ADAC? Either 1) remove ADAC and just test the recent sota baselines and present the work as offline RL in a new problem or 2) justify why we need ADAC. At the moment it is doing the worst of both.
> > >
> > > This is an important point which we should quickly converge on. While we have done this exercise to solve a real world problem, we have not simple applied any $existing$ offline RL method. Our premise is that: (a) Offline RL is practical for TSC, but (b) Existing methods need changes to make them perform well on this problem. So, we indeed need ADAC where locally adaptive penalties are directly motivated by a smoothness assumption that is theoretically shown to be the right choice for the problem. Further, we empirically show how the data collected from cyclic behavioral policy (specific choice stemming from the problem) can be utilized by our adapted offline RL method to perform close to the level of online RL.
> > >
> > > The point first came up because we were requested to try out multiple baselines from recent work. We strongly feel that we have done due diligence in picking two representative model-free and two model-based baseline approaches. (Discrete) TD3+BC already performs very well on a batch generated by partially trained policy and therefore we believe there is not much to gain by including (the discrete action variants of) other contemporary algorithms (CQL, Fisher-BRC, etc.) that have been shown to perform to a similar level on continuous action setting. If the reviewer feels it is important in order to justify need of ADAC, we are ready to add them.
> > >
> > > > I never said [1] was about adaptive conservatism, but it is introduced and therefore not a novel idea so you need to change the framing in your paper. I am fine with ideas being re-used, that is of course natural in science, but do not present ideas as novel when they are not. For clarity, look in Section 6
> > >
> > > Thanks for clearly pointing out to Appendix I of [1]. We will refer to [1] for introducing the idea for adapting penalty hyperparameter. However, we have a different form of adaptive penalty based on local neighborhood built into the reward shaping which does not require post-hoc tuning. So to that extent, we maintain our form of penalty is novel and designed to solve a specific real-world problem.

---

> > > > ### Comment · Reviewer_fEex · 2022-08-22
> > > > **Not sure what you are trying to say**
> > > >
> > > > Addressing the second point first, I think you may have missed that there is no requirement to argue novelty in TMLR - that is the great thing about it!
> > > >
> > > > For the first point, what you said is not at all clear to me. Either 1) Your paper is about showing Offline RL can work on a new problem, so no need to spend so much time justifying a new method. 2) Existing offline RL doesn't work on this problem so you have to introduce this new method. It seems the paper is trying to do both and it makes it much weaker as a contribution.

---

> ### Author Response · Authors · 2022-08-11
> **Response to Review of Paper253 by Reviewer fEex (2)**
>
> > ...it is a huge weakness to not cite a large volume of other works in related areas...
>
> Thanks for pointing this out. We have carried out a major revision of both our technical background and related work section and cited additional relevant work.
>
> > What does the reward shaping term learn? I would expect to see analysis of this given that it is the only change in the paper vs. the previous method (DAC).
>
> We use the maximum reward in the neighborhood as the penalty term — we don’t learn it. As Figure 12 shows, we get out-of-box superior performance than the best DAC model.
>
> > How does this scale to higher dimensional problems, e.g. with multiple similar discrete actions? What about continuous?
>
> (A-)DAC framework is only suitable for discrete action spaces. Size of the action space impacts the size and the diversity required of offline data.
>
> > Minor issues
>
> Thanks for pointing these out. We have fixed them now.
>
> > The model free baselines seem old. There is no CQL, Fisher BRC or IQL. I would expect these to outperform even further on the partial RL dataset.
>
> It should be clarified that the goal of our paper is primarily to solve a real world use case using offline RL. We are not proposing a new general stand-alone algorithm. However we have  made observations  that may be generalizable - like adaptivity of pessimism based on the local reward. Our findings are largely restricted to the particular use case or, at best, can be extrapolated to other discrete action space, continuous state settings. It would, therefore, not be prudent to try the algorithms primarily developed for other settings. We are doing a best-effort analysis by incorporating some of the most popular candidates in our evaluation. TD3+BC is one candidate we have used as a baseline. Our choice is based on its simplicity and its high performance (it happens to be at least on par with the above-mentioned algorithms in the continuous state-continuous action setups). As can be seen from our results, it indeed performs exceedingly well on the partial RL dataset (Figure 9).
>
> > I would hope to see a tuned version of MOPO or for example, MOPO + the same adaptive conservatism bonus...
>
> Incorporating adaptive conservatism in MOPO is not straightforward since our adaptive penalties are based on exploring the state-action space neighborhood. MOPO’s choice for conservatism comes from the prediction of variance given by its ensemble of dynamics models. Nearest neighbor exploration is feasible in finite discrete action setups only that are targeted by our technique.
>
> > Broader Impact Concerns:
>
> Our approach is not “sim-to-data-to-sim” because we start with a  “real” origin-destination matrix that was obtained from local traffic authorities. We use the SUMO simulator to generate routes from “real” data which are consumed by A-DAC to infer a policy. Offline RL has a better chance to be deployed because the current commercial ecosystem is dominated by traffic signal controllers based on classical control methods. Traffic authorities are wary of trying out  “trial and error” approaches -  to generate data, in real settings.

---

### Review · Reviewer_DjJY · 2022-07-29

**Summary Of Contributions:**

This work presents Adaptive(A)-DAC, a model-based offline RL algorithm for traffic signal control. A-DAC is built upon DAC, where the pessimistic reward penalty coefficient is determined adaptively in a way that it penalizes under-explored yet highly rewarding (s,a)-region more. Performance lower bound of A-DAC was provided. In the experiments, A-DAC outperforms offline RL baseline algorithms when the data was collected by either a cyclic policy or partial-RL policy.


**Requested Changes:**

1. In Figure 1, it is a bit unclear to me how A-DAC should be trained faster than DAC since their main difference is just the way the reward penalty coefficient is given. I expected that they should be almost identical in terms of computational cost.
2. In Figure 2, the car is moving "west-to-east", rather than "east-to-west".
3. In the caption of Table 1-2, it would be great to describe the specific meaning of state (x,y): what x is and what x is.
4. page 5: `the DAC-MDP has a finite structure which makes the computation very efficient despite an infinite continuous state space` => I think this can be untrue. DAC-MDP is a non-parametric model, thus it can be computationally very demanding when the number of data points is very large. For example, in Figure 1, we can see that A-DAC is consuming more computation time than MBRL.
5. Page 5: What is the definition of "destination states"?
6. It is unclear how we should deal with the case when NN(s,a,k,alpha) is empty for the newly queried (s,a).
7. Eq (3-4): BCQ and TD3+BC were originally designed for 'continuous action' MDPs, but this paper deals with the problem of discrete action spaces. Thus, it is questionable whether these algorithms are proper baselines for comparison in the experiments. In addition, to make these algorithms work in discrete action space, some modifications seem to be required, but it seems the detailed modifications are not described in the text.
8. Theorem 4.3: DAC paper presents a similar performance lower bound. It would be great to discuss how the bound of A-DAC was improved, compared to DAC's lower bound.


**Strengths And Weaknesses:**

* Strengths
1. This paper aims to present an offline RL algorithm for a real-world problem (traffic signal control), which can be useful for those who deal with this problem.
2. Although novelty is limited due to its nature of slight modification to DAC, the algorithm looks easy to implement and worked well in the empirical study.
3. The paper is generally well-written and easy to follow with some concrete examples.

* Weaknesses
1. There are some unclear parts in the paper. See the requested changes part.
2. The novelty of the algorithm is limited. Adaptive reward penalty looks heuristically designed, rather than theoretically derived.

---

> ### Author Response · Authors · 2022-08-11
> **Response to Review of Paper253 by Reviewer DjJY**
>
> > In Figure 1, it is a bit unclear to me how A-DAC should be trained faster than DAC ...
>
> A-DAC was trained with built-in adaptive reward penalty while the result of DAC is with the best penalty coefficient that was obtained using grid search. This additional evaluation cost for grid search is included in the analysis.
>
> > In Figure 2, the car is moving "west-to-east", rather than "east-to-west".
>
> Please see the revision in Figure 2 caption and in the second para. Section 2.
>
> > In the caption of Table 1-2, it would be great to describe the specific meaning of state (x,y)...
>
> Table 1 and Table 2 captions have been updated now. Additionally, the example in Section 3 where Table 1 is referenced describes the state now.
>
> > page 5: the DAC-MDP has a finite structure which makes the computation very efficient despite an infinite continuous state space => I think this can be untrue. DAC-MDP is a non-parametric model, thus it can be computationally very demanding ...
>
> > What is the definition of "destination states"?
>
> DAC-MDP builds the derived MDP $\tilde{M}$ only on a set of {\em core} states, which is derived from the set of destination states seen in the data. Therefore it has a “finite structure.” The set of core states here is defined as $\cal{S_D} = \\{s'_i | (s_i, a_i, r_i, s'_i)\\}$. Section 3.2 is updated with this information now.
>
> The point on computation being very demanding when the number of data points is large is valid. DAC-MDP manages it to certain extent by using a parallel VI solver. Another way to speed-up computation at the cost of some approximation would be by downsampling input data through clustering first and then build MDP on the clustered data. We did not consider the option here because the scale of data for this particular TSC use case is well supported by the existing computational infrastructure.
>
> > It is unclear how we should deal with the case when NN(s,a,k,alpha) is empty for the newly queried (s,a).
>
> If, for reasonable settings of $k$ and $\alpha$, the nearest neighbors set is empty, it signifies that there is no support in the data set for this query. In this rare event, A-DAC policy selects an action at random. In fact, we never encounter this event in our evaluation; primarily because the evaluation episodes use in-distribution starting states.
>
> This event should act as a guideline for designing additional exploration if the problem setup permits new data collection. (e.g. Suggestion on Slide 5 of “Offline Reinforcement Learning: From Algorithms to Practical Challenges”, NeurIPS 2020 Tutorial)
>
> > Eq (3-4): BCQ and TD3+BC were originally designed for 'continuous action' MDPs, but this paper deals with the problem of discrete action spaces...
>
> Discrete action BCQ is proposed in Section 4 of [Fujimoto et.al., Benchmarking Batch Deep Reinforcement Learning Algorithms]. We describe the same in Section 3.1 and use the code provided by authors in evaluation. We have adapted a couple of state-of-the-art continuous control algorithms, namely TD3+BC as a representative of a model-free algorithm and MoReL and MOPO as representative model-based algorithms, for discrete action control. The modifications required in them are now described in Appendix C.
>
> > Theorem 4.3: DAC paper presents a similar performance lower bound. It would be great to discuss how the bound of A-DAC was improved, compared to DAC's lower bound.
>
> We get a similar theoretical bound but using a less restrictive smoothness assumption, that of Local Lipschitz continuity, on $Q$-function compared to the global Lipshitz continuity in DAC.

---

### Decision · Action_Editors · 2022-09-07

**Recommendation:** Reject

**Comment:**

The paper presents a model-based offline RL solution for a real world traffic signal control. The reviewers found the problem interesting, and the presentation clear. However, the reviewers agree, and I concur, that the paper is not well positioned and its current form not suitable for publication in a machine learning venue.

The paper falls in the category of an applied method (for real-world traffic signal control), that proposed an adaptation of an algorithm (A-DAC over DAC) to solve the problem, and it is not clear who is the main intended audience. I encourage the authors to:

1. Clarify the main positioning of the paper.
2. If the main audience is traffic signal controls community, than the paper would be better suited to a different publication venue.
3. If the main audience is machine learning community, I propose to position paper around the need for adaptive reward penalty under time-variant POMDPs. This would ideally entail finding at least another more commonly used RL environment where there is a need for the adaptive reward penalty, and evaluating the model on it in addition to the traffic signal domain. In addition, under this scenario, the authors should compare the proposed method with RL methods that address overestimation of Q function with regularizes (CQL, Kumar et al. NeurIPS 2020, or DQNReg, Co-Reyes et al., ICLR 2020).